# Highly Sensitive Detection of Benzoyl Peroxide Based on Organoboron Fluorescent Conjugated Polymers

**DOI:** 10.3390/polym11101655

**Published:** 2019-10-11

**Authors:** Mingyuan Yin, Caiyun Zhang, Jing Li, Haijie Li, Qiliang Deng, Shuo Wang

**Affiliations:** 1State Key Laboratory of Food Nutrition and Safety, School of Food Engineering and Biotechnology, College of Chemical Engineering and Materials Science, Tianjin University of Science and Technology, Tianjin 300457, China; mingyuanyinymy@163.com (M.Y.); lijingcgrs@163.com (J.L.); lihaijie198710@163.com (H.L.); 2Tianjin Key Laboratory of Food Science and Health, School of Medicine, Nankai University, Tianjin 300071, China

**Keywords:** fluorescence sensor, benzoyl peroxide, fluorescent conjugated polymer

## Abstract

The method capable of rapid and sensitive detection of benzoyl peroxide (BPO) is necessary and receiving increasing attention. In consideration of the vast signal amplification of fluorescent conjugated polymers (FCPs) for high sensitivity detection and the potential applications of boron-containing materials in the emerging sensing fields, the organoboron FCPs, poly (3-aminophenyl boronic acid) (PABA) is directly synthesized via free-radical polymerization reaction by using the commercially available 3-aminophenyl boronic acid (ABA) as the functional monomer and ammonium persulfate as the initiator. PABA is employed as a fluorescence sensor for sensing of trace BPO based on the formation of charge-transfer complexes between PABA and BPO. The fluorescence emission intensity of PABA demonstrates a negative correlation with the concentration of BPO. And a linear range of 8.26 × 10^−9^ M–8.26 × 10^–4^ M and a limit of detection of 1.06 × 10^–9^ M as well as a good recovery (86.25%–111.38%) of BPO in spiked real samples (wheat flour and antimicrobial agent) are obtained. The proposed sensor provides a promising prospective candidate for the rapid detection and surveillance of BPO.

## 1. Introduction

Benzoyl peroxide (BPO), a versatile substance, has attracted increasing attention, and has been widely employed as an organic initiator, oxidant agent, antibacterial agent, and bleaching agent in industrial production, pharmaceutical formulations, and the food industry [1,2,3,4,5]. However, BPO equipped with strong electron-withdrawing groups is water-insoluble, which makes it a potential environmental contaminant and could be a threat to human health due to the toxicity of peroxide and its various decomposition products (such as benzoic acid, benzoic acid ester, and biphenyl) [6]. BPO could be absorbed into the skin resulting in allergic reactions and its decomposition products, which are difficult to metabolize, usually trigger tissue damage and disease [7,8]. Worse still, BPO is a potential tumor inducer and could cause DNA damage, lipid peroxidation, and thiol oxidation due to its genotoxic effects on human peripheral lymphocytes [9,10]. Considering its potential hazards, reliable BPO analytical methods with high efficiency and accuracy will facilitate the understanding and utilization of BPO, and guiding the clinic treatment.

In recent years, a series of analytical methods for BPO detection, including chromatographic method [11], electrochemistry [9], capillary electrophoresis [2], Raman hyperspectral detection [12,13], and flow injection analysis [14] have appeared in succession. Although these methods are highly specific and sensitive for BPO detection, they still suffer from many problems such as tedious sample pre-treatment, large solvent consumption, expensive instrumentation, and professional operation [15]. Meanwhile, the excellent fluorescent analytical methods based on the small molecular organic compounds as the fluorescence probes have been increasingly employed for sensing of BPO [16,17,18]. Although these analytical methods have inherited the easy-to-manipulate, and great temporal and spatial sampling capability of fluorescence methods, the sensing elements consisting of small molecules are still unable to respond to trace amounts of BPO [10]. Therefore, a new alternative fluorescent analytical method with high signal output, fast spectroscopic response, and feasible measurements should be urgently explored to circumvent this drawback.

Recently, fluorescent conjugated polymers (FCPs), such as polyaniline (PANI), polythiophene (PTPH) and their derivatives, have received significant attention as ideal chemosensing and biosensing materials due to their flexibility in molecular design and prominent signal amplification properties [17,18,19,20,21,22,23,24,25]. Noteworthy, FCPs have a delocalized electronic structure and a large absorption cross-section, which make them attractive as signal transducers in tracing targets even in extremely dilute concentrations [26]. Previous reports have indicated that FCPs had more than 1000-fold amplification than that of their small-molecule model compounds and could enhance the sensitivity of detection [27,28].

Herein, we firstly propose a fast and sensitive fluorescence sensing strategy for BPO based on the organoboron FCPs, poly (3-aminophenyl boronic acid) (PABA). Ordinary PABA as a fluorescence sensor is straightforwardly synthesized by using the commercially available ABA as a functional monomer. The resulting PABA could display sensitively fluorescent response to trace targets due to the inherent ability of boronic acid monomer to form the charge-transfer complexes by combining electron donor/withdrawing groups [29]. For this reason, the fluorescence response led to the establishment of a rapid and highly sensitive PABA fluorescence sensing strategy for the electron-withdrawing BPO in actual samples.

## 2. Materials and Methods 

### 2.1. Materials 

3-Aminophenylboronic acid monohydrate (ABA, 97%), 3-aminobenzoic acid (AMA, 98%), 3-aminobenzene sulfonic acid (ASA, 98%), phenylamine (ANI, 99.5%) and methylbenzene (MB, 99.8%) were obtained from Alfa Aesar Co. Ltd. (Ward Hill, MA, USA). Ammonium persulfate was ordered from Beijing Solarbio Science & Technology Co., Ltd. (Beijing, China). Benzoyl peroxide (BPO) was brought from J&K Chemical (Beijing, China). Hydrogen peroxide (HP 30%) was purchased from Sinopharm Chemical Reagent Co., Ltd. (Shanghai, China). Other reagents were at least of analytical grade and had no further purification. Non-additive wheat flour and an antimicrobial agent were purchased from the local supermarket. Double distilled water (DDW, 18.2 MΩ cm^–1^) was produced by a Water Pro water purification system (Labconco, Kansas City, MS, USA). 

### 2.2. Characterizations of PABA

Fluorescence spectra were obtained on a Shimadzu RF-7600 fluorescence spectrometer (Hitachi, Tokyo, Japan). UV-Vis absorption spectra were obtained using a TU-1901 spectrophotometer (Beijing general instrument, Beijing, China). Fourier transform infrared (FT-IR) spectra (4000–400 cm^−1^) in KBr were recorded in a Vector 22 FT-IR spectrophotometer (Bruker, Karlsruhe, Germany). Morphological characterizations were obtained by scanning electron microscope (SEM) (LEO 1530VP, Carl Zeiss, Oberkochen, Germany) with an attached energy-dispersive X-ray spectroscope (EDS). Thermal characterizations were carried out by STA 449 F5 Jupiter Netzsch thermogravimeter (Netzsch, Selb, Germany). Zeta potentials were measured at room temperature in a neutral water solution with a Zetasizer Nano ZS90 (Malvern, Worcestershire, U.K.). Molecular weights and molecular weight distributions were measured by waters 515 gel permeation chromatography (GPC) equipped with a Waters 2414 differential refractive index detector (set at 35 °C) (Waters, Milford, Massachusetts, USA). 

### 2.3. Synthesis of PABA

PABA was synthesized according to the previous procedure with a slight modification [30]. ABA (2.3 mM), ammonium persulfate (16.0 mM), and deionized water (60.0 mL) were mixed in a round-bottom flask. And then, the mixture was vigorously stirred at 60 °C for 24 h under N2 atmosphere in the dark. The resulting solid was collected by centrifugation, washed with water to remove insoluble inorganic salt and with methanol to purify the final materials. Finally, PABA was obtained by dryness under vacuum conditions. A similar synthetic procedure was employed to prepare the other three FCPs including poly (3-aminobenzoic acid) (PAMA), poly (3-aminobenzene sulfonic acid) (PASA), and poly (phenylamine) (PANI). 

### 2.4. General Procedure for BPO Detection

The stock solution of PABA (0.5 mg mL^−1^) and BPO (8.26×10^−3^ M) were first prepared by dissolving the corresponding target with ethanol. The other concentrations of BPO solution were obtained by diluting the stock solution with ethanol. Then, 500 µL PABA probe solution was taken and mixed with 500 µL BPO sample solution containing a requisite concentration in a test tube, and the mixture was incubated at 50 °C for 5 min in a shaker incubator. The reaction solution was transferred to measure the fluorescence signal with λex/em = 290/370 nm. For comparison, the solution without BPO (as control) was examined under the same conditions. 

For the detection of BPO in wheat flour and gel-like antimicrobial agent samples, the samples were prepared by the following procedure: First, the samples (500.0 mg) were sonicated for 5 min in ethanol (20.0 mL), then the samples were filtered with an organic membrane (pore size: 0.22 μm). The sample solutions were spiked with a requisite concentration of BPO standard solution (0, 0.83, 1.03, 8.26, and 10.30 µM). Finally, the solutions were subjected to the fluorescence analysis following the general procedure given above.

## 3. Results and Discussion

### 3.1. Characterizations of PABA Materials

PABA was facilely synthesized by using the commercially available ABA as a functional monomer and ammonium persulfate as the initiator via a free-radical polymerization reaction (Scheme 1). 

In order to reveal the structural characterizations of the obtained PABA, elemental analyses were first carried out, and the present elementals of C (55.11%), N (8.25%), O (22.09%), and B (14.55%) were observed in PABA (Appendix A), respectively, indicating the successful facile fabrications of PABA. SEM image of PABA (Figure 1A) exhibited a fine agglomerated granular structure which may be attributed to the strong intermolecular H-bonding between PABA chains through –B(OH)_2_ groups [30]. FT-IR spectrum of PABA indicated the absorption at 3267 cm^−1^ arose from stretching vibrations of free –OH in boronic acid, and the peaks observed at 1601, 1345, 1260, and 1075 cm^−1^ were attributed to C=C, B–O, C–N, and B–OH stretching vibration characteristic peaks, respectively. Remarkably, stretching vibrations of –NH in ABA were observed at 3474 and 3389 cm^−1^, and shifted to 3413 cm^−1^ in PABA, which further proved the successful fabrication of PABA (Figure 1B). Thermogravimetric analysis of PABA has demonstrated that the first weight loss around 100 °C arose from the desorption of atmospheric moisture, and the second thermal decomposition observed between 160 and 300 °C were arisen from the dehydrations of boronic acid groups. Final thermal decomposition was started at 300 °C and continued until 560 °C, which was attributed to the decomposition of the backbones. Above 560 °C, thermal decomposition of PABA was observed to slow down, and the residue at 800 °C was determined to be 32%, which indicated that the thermal stabilization of PABA should be considered for sensor materials (Figure 1C). The fluorescence property investigation of PABA indicated that the emission spectrum of PABA has displayed significant solvent-polarity dependence, and the fluorescence emission intensity decreased with increasing polarity of solvents (dimethyl sulfoxide > methanol > acetonitrile > ethanol) (Figure 1D). And the fluorescence excitation and emission wavelength of PABA (290/370 nm) have been confirmed based on the strongest fluorescence emission intensity under different excitation conditions (interval 10 nm) in ethanol (Figure 1E). Meanwhile, the concentration of PABA was closely related to the fluorescence intensity, the fluorescence intensity of PABA increased with the concentration, and the strongest fluorescence emission appeared at 0.25 mg mL^−1^, while the higher concentration lead to the weaker fluorescence emission due to the fluorescent inner filter effect (Figure 1F). Besides, zeta potentials of PABA were further investigated, and the negative values of zeta potentials (−28.4 mV) indicated that the existence of boric acid endowed PABA with strong electronegativity (Appendix A). All these results demonstrated that PABA has been successfully prepared, and had the potential to be a fluorescence sensor.

### 3.2. Sensing of BPO and Mechanism

The ability of boronic acid groups to combine with electron donor or withdrawing groups has accelerated the development of organoboron sensors materials [29]. Considering the strong electron-withdrawing groups of BPO and the outstanding performance of boronic acid groups in tracing amounts of targets, PABA might be an ideal fluorescent probe for the sensing of BPO. As shown in Figure 2A, the fluorescence emission of PABA could be significantly quenched by BPO. Herein, we speculate that the general concept lies in the fact that the formation of charge-transfer complexes between PABA and BPO in organic solvents quenches fluorescence emission. To check this hypothesis, UV-vis absorption spectrum of PABA treated with BPO was investigated, and a newly generated absorption band III; was obviously visible (Figure 2B), meanwhile the perturbation of the absorption spectrum of PABA was observed with the increasing of BPO concentration (the spectral band was broadened, the absorption intensity was heightened and the solution color was deepened) (Appendix A), which verified the formation of charge-transfer complexes and also demonstrated a static quenching feature. In addition, the fluorescence quenching was unrelated to the fluorescent inner filter effect due to the absence of complete overlapping between the BPO absorption (270 nm) with the fluorescence excitation and emission wavelength of PABA (Appendix A), and the slight change of molecular weight before/after PABA treated with BPO also showed that the continuous polymerization reaction did not occur to trigger the fluorescence quenching (Appendix A), which further confirmed our assumption. It is well known that the higher temperature will typically result in the dissociation of weakly bound complexes, and hence smaller amounts of static quenching [31]. In order to further verify the static quenching, the comparison of the fluorescence quenching rate of PABA treated with BPO at 20 °C and 50 °C has been carried out, and the result did exhibit the typically static quenching due to the higher fluorescence quenching rate at low temperature (20 °C) than that at the high temperature (50 °C) (Figure 2C).

In order to further verify the performance of boronic acid groups in PABA for sensing of BPO, three FCPs (PAIN, PASA, and PAMA) were synthesized from different functional monomers (ANI, ASA, and AMA), and characterized by elemental analyses, SEM images and FT-IR spectra, respectively (Appendix A). And the investigations of the fluorescence properties and zeta potentials of the three FCPs have also shown significant differences due to the introduction of variant functional monomers (Appendix A). When the same concentration BPO was added, the fluorescence of all the four FCPs could be quenched (Figure 3A). Considering the fact that the fluorescence quenching of FCPs was unrelated with the fluorescent inner filter effect (Appendix A), the alternate backbone of benzene and amino groups in the four FCPs could cause some degree of the fluorescence quenching due to electrostatic and hydrophobic interaction with BPO. Significantly, the fluorescence response of the four FCPs to the same concentration BPO was quite different, the fluorescence quenching degree declined in the order of PABA > PANI > PAMA > PASA, which could confirm that functional groups play an important role. In fact, PANI has no additional functional groups on main backbone structure, while PAMA and PASA have the electron-withdrawing carboxyl groups (–COOH) and sulfonic acid groups (–SO_3_H) onto the backbone structure, respectively [32,33], which could reject the interaction with BPO and lead to the weaker fluorescence quenching than that of PANI. In contrast, PABA contained boronic acid groups displayed the strongest fluorescence quenching among these materials, which was attributed to the transformation ability of the boronic acid group from the sp^2^-hybridized atom to the sp^3^-hybridized atom, acting as electron donor groups to reinforce the interaction between PABA and BPO [29,34]. In order to further explore the interaction between PABA and BPO, the relative fluorescence intensity of PABA titrated with HP (containing peroxyl radical) and MB (containing benzyl ring structure) were investigated. As shown in Figure 3B, the selected two targets could cause the fluorescence quenching due to the hydrophobic interaction between benzyl ring in MB and PABA, and the interaction between peroxyl radical in HP and PABA. However, BPO showed the strongest fluorescence quenching to PABA, which was attributed to the synergistic effect of peroxyl radical and benzyl ring in BPO. Based on these results, we reasoned that the strong fluorescence signal response of PABA towards BPO was mainly attributed to the formation of charge-transfer complexes between PABA with BPO where boronic acid groups made a crucial role (Scheme 1B). Therefore, PABA could be a promising sensor to realize the sensing of BPO.

### 3.3. Optimization of Experimental Conditions

In order to achieve superior sensitivity, assay parameters, such as the concentration of PABA, the response time, and the reaction temperature, were investigated. Considering the fluorescence emission of PABA and the solubility of BPO, the following experiments were carried in ethanol media. The fluorescence response of different concentration PABA towards PBO (4.13 × 10^−3^ M) was first examined, revealing that 0.25 mg mL^−1^ of PABA has significant fluorescence quenching ability (Figure 4A). Thus, such concentration was adopted. The time-dependent fluorescence response of PABA towards BPO showed that the fluorescence intensity of PABA at 370 nm was in almost constant level with the increasing time, indicating that the fluorescence response was fast and stable, which met the rapid detection requirements (Figure 4B). And considering the need of detection, 5 min was chosen as the optimum response time. In addition, the reaction temperature was crucial to a practical optical sensor. The fluorescence response of PABA towards BPO enhanced with the temperature increased from 20 to 50 °C attributing to the strong charge transfer effect of PABA, while higher temperature caused a weaker fluorescence response due to the volatilization of ethanol (Figure 4C). Therefore, 50 °C was confirmed as the optimum temperature. On the basis of the above observations, the reaction of PABA (0.25 mg mL^−1^) with BPO at 50 °C for 5 min in ethanol was confirmed as the optimum analytical condition.

### 3.4. Calibration Curve

In order to evaluate the limit of detection (LOD) of the proposed method for BPO, we plotted the calibration curve by adding different BPO concentrations and monitoring the change in fluorescence intensity at 370 nm. We observed that the fluorescence intensity of PABA gradually decreased with increasing BPO concentration (Figure 5A) and the negative logarithm of BPO concentration was proportional to the quenching in PABA fluorescence intensity (F_0_–F_i_) with a linear relationship of Y = −198.94 X + 1819.44 (R^2^ = 0.9941) in the detection range of 8.26 × 10^−9^ M – 8.26 × 10^−4^ M (Figure 5B). The LOD is usually obtained from three times of signals to noise (S/N = 3) [8,35,36,37,38] and the estimated LOD was 1.06 × 10^−9^ M, featuring a high sensitivity. With the developed method, the ultrasensitive detection for BPO can be completed within a short time. In addition, the sensitivity of this method was higher than that of previous fluorescence methods [1,10,37,39] highlighting its potential use for BPO detection.

### 3.5. Resistance to Interference Substances

In order to further investigate the selectivity of this proposed method for BPO, we examined the fluorescence responses of PABA towards various interference substances, which might coexist in food and an antibacterial agent. As shown in Appendix A, when PABA was treated with equal concentration common ions (8.26 × 10^−6^ M) (anions: F^–^, Cl^−^, Br^−^, I^−^, NO_3_^−^, H_2_PO_4_^−^, HCO_3_^−^, and SO_4_^2−^ and metal ions: Ni^2+^, Mg^2+^, Cu^2+^, Zn^2+^, Co^2+^, Fe^2+^, Ba^2+^, and Ca^2+^), no obvious fluorescence response was observed. Meanwhile, the fluorescence quenching of PABA triggered by amino acids (l-glycine, l-cysteine, l-tyrosine, and l-proline) and saccharides (d-mannose, d-fructose, d-glucose, Maltose, Sucrose, and Amylum) was also slight. Although ascorbic acid and H_2_O_2_ caused certain fluorescent quenching, these have little influence on the selectivity of PABA towards BPO due to their easy degradation. Herein, we concluded that PABA had strong resistance against other possible interference existing in a complicated environment.

### 3.6. Analytical Applications

BPO has been widely used as a food additive and an antibacterial agent, while the existence of excess BPO residues could be a serious threat to human health. Therefore, the accurate assessment for BPO was of great significance. In view of the excellent sensitivity and selectivity of the PABA sensing strategy for BPO, the practical application for tracing BPO in real samples (wheat flour and an antimicrobial agent) was further investigated. The processed real samples were first spiked with different amounts of BPO standard solution (0, 8, 10, 80, 100 mg kg^−1^ suggesting a requisite concentration of BPO existed in these samples), and then the fluorescence intensity of PABA treated with these real samples was recorded under the ascertained optimized detecting conditions. And the levels of BPO in these samples were further determined according to the standard calibration curve. The result showed good recovery values towards BPO (89.37%–111.38% in wheat flour and 86.25%–107.54% in an antimicrobial agent, Table 1). Meanwhile, the result showed that the proposed sensor could determine BPO at a low concentration level (8 mg kg^–1^) in wheat flour, which was below the level of BPO (75 mg kg^–1^) in the Codex Alimentarius Commission standard [36]. Therefore, this method has exhibited a good capacity to quantify BPO in real samples.

## 4. Conclusions

In this study, a novel fluorescent strategy was established for tracing of BPO based on the facile synthesized organoboron FCPs PABA. The sensing mechanism is mainly depended on the formation of charge-transfer complexes between PABA and BPO. The proposed strategy provided a broad linear range and an excellent sensitivity compared to the previously reported approaches (Appendix A). The practical application was demonstrated by sensing trace BPO in real samples (wheat flour and an antimicrobial agent) with a good recovery. Thus, our research provides a simple, rapid, and effective approach for the quantification of BPO. The proposed PABA fluorescent strategy may be a promising application for the sensing of BPO in more complex samples.

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
