# Peer review of "Highly Sensitive Detection of Benzoyl Peroxide Based on Organoboron Fluorescent Conjugated Polymers"

_polymers, 2019, doi:10.3390/polym11101655_

Round 1

Reviewer 1 Report

Wang and co-workers report a fluorescent strategy for the rapid quantification of BPO traces based on the use of organoboron-containing polyaniline polymers. The manuscript is well written and the results are clearly presented. The characterization of synthesized polymers was seriously performed by using common techniques. I however have some questions concerning TGA analysis: 1) It was performed under He/Ar or under air? What is the nature of the 32% material at the end of the analysis? 2) How the weight losses at 100°C and between 160° and 300°C were attributed? Did the author use TGA coupled with MS analysis?

I therefore suggest acceptance after minor revision

Author Response

Some questions concerning TGA analysis: 1) It was performed under He/Ar or under air? What is the nature of the 32% material at the end of the analysis? 2) How the weight losses at 100°C and between 160° and 300°C were attributed? Did the author use TGA coupled with MS analysis?

Answer: Thank you very much for your kind suggestion. In order to test thermostability of PABA, TGA analysis was carried out, which was not coupled with MS analysis. The process was performed under N2, and the nature of the 32% material at the end of the analysis could be the residue of the decomposition of polyaniline back bones. The first weight loss around 100 °C could be arisen from the desorption of atmospheric moisture, and the second thermal decomposition observed between 160 – 300 °C could be arisen from the dehydrations of boronic acid groups. The interpretations above were referred to the previous literature. [1]

References

Gumus, O.Y.; Ozkan, S.; Unal, H.I. A Comparative Study on Electrokinetic Properties of Boronic Acid Derivative Polymers in Aqueous and Nonaqueous Media. Macromolecular Chemistry and Physics 2016, 217, 1411-1421, doi:10.1002/macp.201500524.

Reviewer 2 Report

Yin, M. et al. have reported a novel PABA that detects benzoyl peroxide through the fluorescence-quenching mechanism. Appropriate control experiments have been carried out, which indicate the probe is superior compared to PANI, PAMA and PASA. Its practical utility has also been demonstrated.

Minor comments

F0 and Fi need to be defined explicitly (F0 and Fi are the fluorescent intensity of PABA in the absence and presence of BPO, respectively?) in Figure 2 legend as well as in Figure S6 legend. Please cite the following paper in the introduction part discussing about small molecular organic compounds Commun., 2012,48, 2809-2811 by Huimin Ma and co-worker. Line 271: it should be NO3- Figure 4 and its legend do not match with each other. (for example, Figure 4B is related to response time, but it is written as reaction temperature. Line 63: Straightforward---Straightforwardly; line 33: could threat—could be a threat to etc.

Author Response

1. F0 and Fi need to be defined explicitly (F0 and Fi are the fluorescent intensity of PABA in the absence and presence of BPO, respectively?) in Figure 2 legend as well as in Figure S6 legend.

Answer: Thank you very much for your kind suggestion. We have checked the definition carefully and changed the sentence “F0 and Fi are the fluorescent intensity of PABA in the presence and absence of BPO” into “F0 and Fi are the fluorescent intensity of PABA in the absence and presence of BPO, respectively” in Figure 2 legend and changed the sentence “ F0 and Fi are the fluorescent intensity of PABA in the presence and absence of different substance” into “F0 and Fi are the fluorescent intensity of PABA in the absence and presence of different substance, respectively” in Figure S6 legend.

2. Please cite the following paper in the introduction part discussing about small molecular organic compounds Commun., 2012,48, 2809-2811 by Huimin Ma and co-worker.

Answer: Thank you very much for your kind suggestion. The paper of Commun., 2012,48, 2809-2811 by Huimin Ma and co-worker has been cited in reference [16].

3. Line 271: it should be NO3- Figure 4 and its legend do not match with each other. (for example, Figure 4B is related to response time, but it is written as reaction temperature. Line 63: Straightforward---Straightforwardly; line 33: could threat—could be a threat to etc.

Answer: Thank you very much for your kind suggestion. We have checked the manuscripts carefully. The mistakes in line 271 and Figure 4B have been corrected. And the word “Straightforward” has been changed into “Straightforwardly” in line 63 and “could threat” has been changed into “could be a threat to” in the line 33.
